# Cardiac Rehabilitation in Primary Care. Impact of an Intervention on Perceived Self-Efficacy

**DOI:** 10.3390/jfmk4020032

**Published:** 2019-06-01

**Authors:** Antonio J. Madueño-Caro, Manuel L. Mellado-Fernández, Manuel Pardos-Lafarga, Marta Muñoz-Ayllón, Laura Saez-Garcia, Juan Gómez-Salgado

**Affiliations:** 1La Laguna Health Centre, Andalusian Health Service, 11010 Cádiz, Spain; 2Casines Health Centre, Andalusian Health Service, Puerto Real, 11010 Cádiz, Spain; 3Department of Sociology, Social Work and Public Health, University of Huelva, 21007 Huelva, Spain; 4Safety and Health Posgrade Program, Universidad Espíritu Santo, Guayaquil 091650, Ecuador

**Keywords:** cardiac rehabilitation, primary care, self-efficacy, patient safety, health care quality, access, evaluation

## Abstract

Cardiac rehabilitation is cost-effective and should be considered a part of the care system provided to patients who have suffered a myocardial infarction or another heart disease. The main variable to study was the scoring, prior to and after the intervention in the General Scale of Self-Efficacy by Baessler & Schwarzer. A clinical community trial that was open controlled and randomised was used. All adult subjects of both sexes who had completed a cardiac rehabilitation program for 12 months at the reference hospital were selected and offered to participate. The psychometric variables registered were the Salamanca screening questionnaire, the Hamilton Anxiety Rating Scale, and the Beck Depression Inventory. Eighty-nine subjects accepted to participate in the study (93.89% response rate), with an average age of 63.01 years (SD 8.75). Once the study was concluded, the main outcome was a difference in means of 6.09 points in the General Scale of Self-Efficacy (*p* < 0.0053, 96% confidence interval—4.1950–10.29), showing that the group exposed to the intervention reached a higher score in the above-mentioned scale. However, there were no significant differences (*t*-student 0.1211; *p* = 0.943) after the estimation and contrast of population means for score differences between the groups regarding the Hamilton scale. Similarly, there were no significant differences between the groups regarding the means obtained in the variable score difference in the Beck Depression Inventory (*t*-student −0.1281; *p* = 0.8987). The results showed an increase in those scores related to general self-efficacy among the population that completed the intervention program, as compared to the control group.

## 1. Introduction

Cardiovascular diseases are the leading cause of death in Spain. There are independent cardiovascular risk factors (hypertension, dyslipidaemia, sedentary lifestyle, smoking, and stress), and some of them are modifiable. Their incidence can be reduced by facilitating reinsertion into daily life through secondary prevention, lifestyle modification, and physical and psychological rehabilitation [1].

The World Health Organization (WHO) defined Cardiac Rehabilitation (CR) as the sum of all the necessary measures to give the post-myocardial infarction (post-MI) patient the best physical, psychological, and social conditions to recover a normal position in society and a life that is as productive and active as possible (WHO, 1993). CR is cost-effective and should be considered a part of the care system provided to patients who have suffered a myocardial infarction or another heart disease [1,2,3,4,5,6]. It has been proven to reduce mortality in people with a coronary disease or, at least, to improve their quality of life [7,8,9,10,11,12].

For many years, primary care (PC) has devoted time and effort to patients’ training on primary prevention. Currently, more dedication is required for secondary prevention due to the volume of patients diagnosed with coronary diseases that attend health centres, something derived from the increased longevity and survival rate after acute myocardial infarctions and other cardiovascular events.

CR is classified in three stages. The first one (phase I) starts at the bedside of the hospitalised patient after suffering a cardiac event. It encompasses basic information and activity planning. The so-called phase II [13,14] takes place after hospital discharge, but the patient remains under direct supervision of the cardiology service through nursing and rehabilitation actions, in addition to the strictly cardiology ones. The role of nursing is essential and has been evidenced from experience, both in primary prevention education and in the development of self-care theories through nursing processes for patients with cardiovascular risk factors.

The maintenance of cardiac rehabilitation activities for these patients throughout their lives is called phase III, and is also known as a prolonged or long-term cardiac rehabilitation. At this moment, the continuity of the care process, the recommendations, and the activities planned for the future must be ensured. The role is mainly developed at the PC level and is an adequate way to control and detect cardiovascular risk factors, reducing the morbidity these diseases originate. In Spain, outpatient CR implementation is just anecdotal [15,16].

Despite the existing evidence, there is little development of hospital CR units in primary care environments (phase II of CR). [13,14,15,16,17,18,19]. For this reason, most patients are sent back home without a start-up program that advises them on their illnesses and the changes in habits that are fundamental for their recovery. This phase has the peculiarity that the person may have already returned to his/her routine activities and/or to the workplace, and this phase extends in time for the rest of the patient’s life cycle. [19,20].

An intervention aimed at the patient’s physical condition may not be enough, as the MI significantly affects the patients’ living conditions in practically all areas of their daily activities. Psychological factors play a greater role in the course and treatment of CR. Patients actively develop models that are based on their disease, treatment, and recovery, thus having an influence on their behaviour and emotions [21,22,23,24,25].

On the other hand, ‘perceived self-efficacy’ refers to people’s beliefs about their own abilities to achieve certain outcomes [4]. The ‘self-efficacy’ term has been applied to various behaviours that affect different health domains (chronic diseases, drug use, sexual activity, smoking, physical exercise, weight loss, and also the ability to recover from health problems or to avoid potential health risks). The main measurement tool referenced is the General Self-Efficacy Scale by Baeesler and Schwarzer [26]. Several investigations have revealed that high levels of perceived self-efficacy have beneficial consequences for the individual’s functioning and general well-being [27].

Therefore, in order to achieve an effective CR and a lasting effect on the patient, it would be useful to introduce psychotherapeutic assistance and interventions that increase the perception of self-efficacy, in addition to the physical therapeutic intervention. This, combined with adequate adherence to healthy behaviours that may decrease the risk of subsequent cardiac events, will be beneficial in the rehabilitation process.

In our country, therapeutic objectives have been proposed for the reduction of the emotional impact caused by myocardial infarction, as well as for the reduction of the probability of subsequent myocardial events [28,29].

Consequently, the objective of this study is to identify the self-efficacy level regarding perceived health among patients who have suffered a cardiac event and have been through a hospital cardiac rehabilitation phase. In addition, we aim at verifying whether the patients’ inclusion in a primary health care cardiac rehabilitation program allows an increase in the level of perceived self-efficacy.

## 2. Materials and Methods

### 2.1. Design

Community clinical trial; open; controlled and randomised study.

### 2.2. Field of Study

Urban Centre of Primary Care. All the subjects who compose the study cohort attend this centre, although these subjects are related to ten different health centres that belong to the hospital area of reference.

### 2.3. Participants and Study Duration

All participants (adults of both sexes) who concluded phase II in the CR unit of the reference hospital were offered to participate over 12 months, so it was not necessary to establish a sample size.

### 2.4. Sample Size, Sampling, and Randomisation Procedure

The study population is composed of the total number of discharged patients from the cardiac rehabilitation unit of the Puerta del Mar Hospital during a calendar year (June 2013 to June 2014), which is a total of 95 people.

To determine the sample size, the research team used the General Self-Efficacy Scale by Baessler and Schwarzer [26].

The specific formula for calculating a sample size in the case of a comparison of two means in a bilateral test was applied. As outcomes, the mean’s standard deviation of the main dependent variable was 12% (perceived general self-efficacy), and the means difference between the groups, which is considered of clinical relevance, was of at least 6 points in the aforementioned General Scale of Self-Efficacy.

A precision of 95% and a test power of 80% are accepted. The result proposes a sample of 44 subjects in each group. As the study population is made up of the total number of patients discharged from the cardiac rehabilitation unit of the Puerta del Mar Hospital, 104 people participated during a calendar year, and all the subjects who met the feasibility criteria were offered to participate in the study. Thus, for 12 months, all subjects of legal age, both sexes, and who had concluded phase II in the CR unit of the university hospital Puerta del Mar (Cádiz) were offered to participate in the study. When discharged from phase II of the CR in the cardiac rehabilitation unit, the patients were immediately informed of the study, asked for their participation, and offered an informed written consent. Randomly and blindly (a closed envelope given to the researcher), the subjects were assigned either to the intervention or the control group, using a pre-established 2:1 ratio. A non-response or refusal registration to participate in the study was performed for the non-response analysis.

### 2.5. Exclusion Criteria

Severe mental illness, difficulty in understanding or using the Spanish language, not belonging to the health area where the study is developed, or impossibility to travel to the health centre were the exclusion criteria.

The research was carried out in accordance with the 1975 Declaration of Helsinki (revised in 2008). According to point 23 of the Declaration, an approval by an ethics committee was obtained. The research project protocol has been approved by the ‘Comité de Ética e Investigación del Hospital Universitario Puerta del Mar y Distrito Bahía de Cádiz La Janda’, with approval number ‘13/13′ (26 March 2013).

All participants gave their informed consent for the research, and were ensured that their anonymity was preserved.

### 2.6. Variables

Dependent primary efficacy variable: Prior to intervention and post-intervention scores in the Baessler & Schwarzer’s General Self-Efficacy Scale [26] (Figure 1).

Universal and sociodemographic variables were determined: Age, sex, body mass index (BMI), stratification of cardiac risk at discharge from the hospital CR phase, and reference health centre by town.

Psychometric variables: Salamanca screening questionnaire [30], Hamilton Anxiety Rating Scale [31,32], and Beck Depression Inventory [33].

All these scales, questionnaire or inventory, have been validated in the Spanish population and language [26,29,30,34,35,36].

### 2.7. Intervention

The intervention was performed by a succession of family physician visits on a scheduled basis.

### 2.8. Intervention Group

Visit 0. (Hospital rehabilitation unit, research physician, individual visit). Delivery and reading of information sheet. Signing of an informed consent. Registering telephone number and electronic address. Determination of baseline variables.

Visit 1. (Health centre, family physician, individual visit). Prior to 30 days after discharge from hospital CR unit. Individually-adapted planning (recommendations on medical treatment and adherence, adaptation of physical and nutritional activity plan, as well as recommendations to optimise the control of present risk factors and of healthy lifestyles).

Visit 2. (Health centre, family physician, individual visit). Three months following the start of phase III. Brief cognitive intervention in a clinical setting mediated by a motivational interview, detecting possible cognitive distortions in relation to the disease and the working environment, relationships, partner, and sexual, physical, and nutritional activities.

Visit 3. (Health centre, family physician, group visit, 6–8 subjects). Six months following the start of phase III. Reminder session on the physical activity plan learnt during the hospital rehabilitation phase. It is attended in sports clothes.

Visit 4. (Health centre, family physician, group visit, 6–8 subjects). Nine months following the start of phase III. Content related to the concepts of rehabilitation, self-care, and resilience. Qualitative and participatory methodology:
Strategies to recognise and work against resistance to change (arguing, interrupting, denying, ignoring).Encourage patients to evaluate their own behaviour and accept new challenges.Encourage patients to be able to control their diseases’ progression.


Visit 5. (Health centre, research physician, individual visit). Determination of variables.

### 2.9. Control Group

Visit 0 (Hospital Rehabilitation Unit, research physician, individual visit). Delivery and reading of information sheet. Signing of informed consent. Determination of baseline variables.

Patients are placed under the standard care of their area family physician, community nursing, and cardiology service.

Visit 1 and conclusion of visits. 12 months following the discharge from the hospital Cardiac Rehabilitation Unit. Determination of variables.

### 2.10. Statistical Analysis

Descriptive statistics are performed: distribution of absolute and relative frequencies for categorical variables, and measuring of central tendency and dispersion for quantitative variables. Homogeneity is determined for the variables between the intervention and control groups at the baseline visit. Determination of bivariate analysis for the variables between the groups, as well as hypothesis contrast estimation through the student’s *t*-test determination, seeking means differences between groups. Statistical analysis licensed for SPSS version 15.0 for Windows (Copyright © SPSS Inc., Redmond, WA, USA, 1989–2006).

## 3. Results

Ninety-five subjects out of a total of 104 selected patients agreed to participate, of which 89 completed the study period (Figure 2) and were included in the analysis (93.68% response rate), with a mean age of 63.01 years (SD 8.75). Regarding their sex, out of the total, 80.23% were male and 19.77% were female. After the randomisation following a 2:1 ratio, 58 subjects were assigned to the intervention group (61.05%) and 37 to the control group (38.95%).

The data collected during the baseline visit allowed us to determine the homogeneity between the intervention and control groups for the variables: age, diagnosis, risk stratification, anxiety scores, depression, and perceived self-efficacy, once the random assignment to the groups was made (Table 1).

The perceived self-efficacy was inversely correlated with the scores obtained in the Hamilton Anxiety Questionnaire (Pearson *r* = −0.04009, *p* = 0.0018) and in the Beck Depression Inventory (*r* = −0.4152, *p* = 0.0012). Similarly, the perceived self-efficacy score was inversely correlated with the scores obtained for dependent (Pearson *r* = −0.3175, *p* = 0.0152), impulsive, unstable, antisocial, or schizotypal personality traits. The correlations were not relevant for other personality traits (Table 2).

The intervention group participants who attended all the program sessions [5], and those from both groups who completed the Self-Efficacy Scale at the beginning and at the end of the study (n = 50), were included in the outcomes analysis related to the main dependent variable. To obtain the difference of mean scores between groups, regarding the ‘perceived general self-efficacy’ variable, as well as for the variables related to anxiety and depression scores, new variables are coded, called the ‘score difference in perceived general self-efficacy’, ‘difference in Hamilton Anxiety Rating Scale score’, and ‘score difference in Beck Depression Inventory’. This is the result from the subtraction of the scores obtained from each participant upon completion of the General Self-Efficacy Scale at the beginning and at the end of the study. This scale offers a range of punctuation between 0 and 40, depending on the individual’s response to each of the 10 items that compose it. Thus, in each question, it can be answered by choosing between options “incorrect” (punctuation 1), “barely certain” (2), “rather true” (3), or “true” (4). The maximum score, therefore, is 40 points.

Following the end of the study, and as its main outcome, a difference of 6.0972 points (*p* 0.0053, 95% CI 4.1950 to 10.29) was detected after estimation and contrast of hypotheses for population means between groups, with the intervention group being the one that showed a higher score improvement (Table 3).

However, in the final visit, there were no significant differences (*t*-student 0.1211; *p* = 0.943) after the estimation and contrast of population means for score differences between the groups regarding the Hamilton scale. Similarly, there were no significant differences between the groups regarding the means obtained in the variable ‘score difference in Beck Depression Inventory’ *t*-student −0.1281; *p* = 0.8987).

## 4. Discussion

Cardiac rehabilitation has proven to be efficient in terms of patients’ morbidity and mortality [1]. Likewise, an increase in perceived self-efficacy improves the functioning and well-being of the patients. In spite of this evidence, there are few cardiac rehabilitation programs in primary care, especially in Spain [17,20].

When interpreting the results of this study, there is a tendency towards the improvement of the feeling of perceived general self-efficacy regarding their health in those patients who perform systematised intervention programs in the primary care setting. In addition, the results are consistent with the bibliographic evidence. In fact, in cardiac rehabilitation programs, an emphasis on motivational factors affects the patient’s adherence. It is suggested that in cardiac rehabilitation programs more attention should be paid to the role of group-based rehabilitation. Psychoeducation resulted in improved mental health in patients with coronary heart disease [6]. There is evidence about benefits in this approach when performed as a complementary, effective, non-invasive nursing intervention to reduce psychological problems in these patients [1]. There are results that also suggest that cardiac rehabilitation programs that were successful on proximal targets (systolic blood pressure, smoking behaviour, physical exercise, and emotional distress) were more effective on distal targets (cardiac mortality and MI recurrences) than programs without the success of proximal targets [23].

However, there are studies that investigated the effectiveness of an exercise programme in patients discharged home following a critical illness by measuring the quality of life test (SF-36), and the improvements in the primary outcome did not differ significantly between studied groups [37].

The results of this study are not consistent with data about women’s participation or dropout from cardiac rehabilitation programs [4].

Considering the results, there is a trend towards the inverse association between perceived self-efficacy and mood among the population who suffer a cardiac event, as well as in relation with certain personality traits. This suggests that personality traits may play a role in the detrimental effect of emotional distress in MI patients [23]. However, there is no improvement after participating in the intervention program of cardiac rehabilitation phase III in the score regarding symptoms of anxiety or depression. This apparent contradiction, in the opinion of the research team, could be explained, or at least subjected to debate, by the concept of ‘resilience’, recently incorporated into the psychology of health. Suniya Luthar, between 2000 and 2006, defined ‘resilience’ as a positive adaptation despite adversity, emphasising the two elements that constitute the term: adversity and adaptation.

The present study has met the recommendations for improving the quality of reporting of randomised controlled trials contained in the CONSORT Statement [38].

### Limitations

As a limitation, the sample size is considered to be too close to the precise calculation. Even though statistical significance that meets the clinical relevance existing in the difference of means regarding the main variable has been found, the inclusion of a larger population size is seen as an opportunity.

Another limitation is the lack of determination of correlations between the attainment of objectives related to the control of cardiovascular risk factors and the overall self-efficacy score. Despite this, it is worth highlighting the prominence and the opportunity to measure the perceived general self-efficacy of an individual who has suffered a cardiac event as an objectification of the feeling of the patient’s ability to improve their own state of health beyond the specific results that can be obtained from physiological variables.

A patient’s resilient attitude may find plausibility and coherence with a feeling of perceived self-efficacy for one’s own health care, even though there is no improvement in the scores related to anxiety symptoms or sadness. The person’s coping and adaptation ability may integrate symptoms which are not related to joyful, euphoric, or happy moods. This fact could explain the results of a lack of improvement in scores regarding the scales that represent the affective state despite obtaining benefits for the feeling of self-efficacy.

## 5. Conclusions

The results of the present study show significant improvements in the scores related to the feeling of perceived general self-efficacy for the population that concluded the intervention program in primary health, as compared to the control group.

Perceived self-efficacy is inversely correlated with the scale scores that measure anxiety or depression symptoms. The improvement in the scores regarding the feeling of self-efficacy perceived for one’s own health is accompanied by an improvement in those scores related to the patient’s state of mind.

Further studies on the field are considered pertinent, including using a larger sample size and performing the study in different health areas. We should be able to test the effectiveness of systematised interventions adapted to feasibility criteria within the health system. The interventions should be aimed at increasing perceived self-efficacy among similar populations to the studied one.

## Figures and Tables

**Figure 1 jfmk-04-00032-f001:**
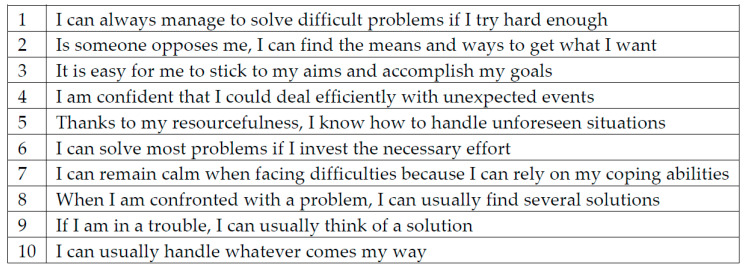
General Self-Efficacy Scale by Baessler and Schwarzer.

**Figure 2 jfmk-04-00032-f002:**
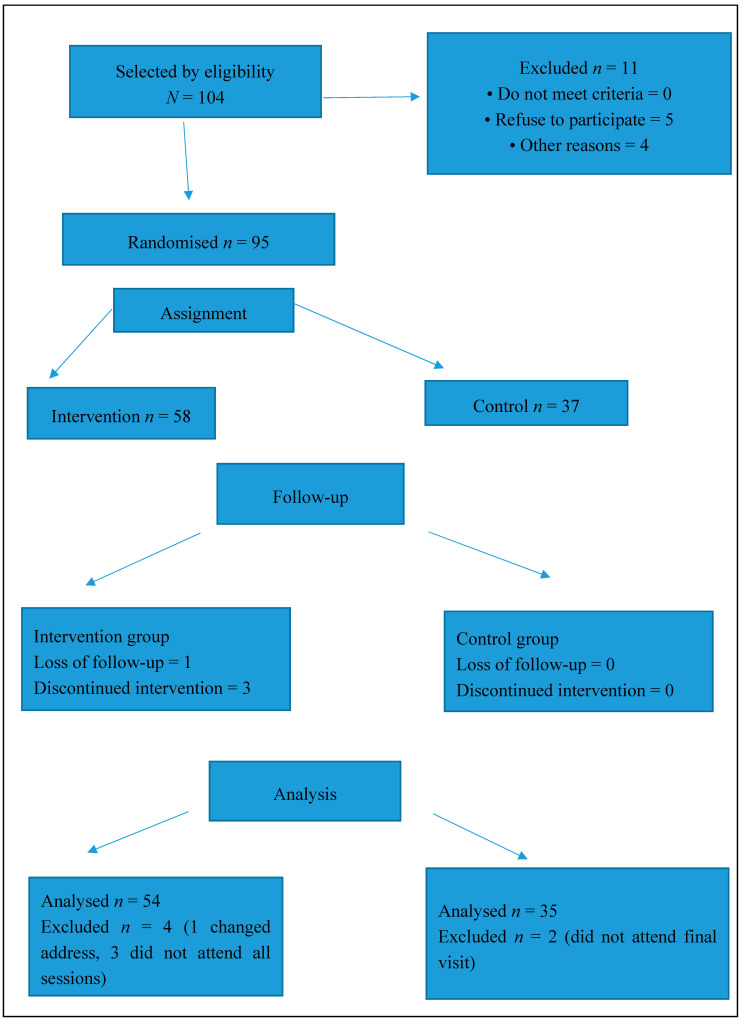
Eligibility, randomisation, and follow-up.

**Table 1 jfmk-04-00032-t001:** Baseline data for the variables. No significant differences between groups.

Variable	Control Group %	Intervention Group %	Differences
Age	63.34 (DE 9.2)	62.94 (DE 8.52)	(*p* 0.83)
Diagnosis			(*p* 0.5741)
Monovascular disease	11.24	11.24	
Polyvascular disease	5.62	12.36	
Non-ST elevation Coronary Syndrome	5.62	14.61	
Acute myocardial infarction	7.87	14.61	
Risk stratification			(*p* 0.3075)
Low	20.45	35.23	
Moderate	10.23	20.45	
Severe	7.95	5.68	
Anxiety score (average)	15.52 (DE 11.2)	13.32 (DE 11.56)	(*p* 0.4482)
Depression score (average)	9.68 (DE 7.07)	9.52 (DE 9.4)	(*p* 0.9369)
Perceived self-efficacy (average)	27.77 (DE 7.03)	27.34 (DE 5.8)	(*p* 0.7761)

**Table 2 jfmk-04-00032-t002:** Pearson correlations between perceived self-efficacy, personality traits, and affective state at baseline visit.

		Self-Efficacy
Personality trait	Dependent	*r* = −0.3175 *p* = 0.0152
	Impulsive Unstable	*r* = −0.4243 *p* = 0.0009
	Borderline	*r* = −0.2841 *p* = 0.0307
	Antisocial	*r* = −0.2829 *p* = 0.0314
	Schizotypal	*r* = −0.2831 *p* = 0.0313
	Paranoid	*r* = −0.1321 *p* = 0.3185
Affective state score	Anxiety Severity (Hamilton)	*r* = −0.4009 *p* = 0.0018
	Depression Inventory (Beck)	*r* = −0.4152 *p* = 0.0012

**Table 3 jfmk-04-00032-t003:** Hypothesis test for means difference between groups.

Group	Intervention	Control
Sample	54	35
General Self-Efficacy Scale means differences between Baseline to final visit punctuations	3.3750	−2.7222
Standard deviation	7.0196	7.1932
Means punctuation difference	6.0972 IC (4.1950 to 10.29); *p* 0.0053	

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
