# Peer review of "Cardiac Rehabilitation in Primary Care. Impact of an Intervention on Perceived Self-Efficacy"

_jfmk, 2019, doi:10.3390/jfmk4020032_

Round 1

Reviewer 1 Report

Cardiac rehabilitation is an indispensable part of the health care system  provided to patients who have suffered a myocardial infarction or another heart disease. The objective of the study was to identify the self-efficacy level regarding perceived  health among patients who  suffered a cardiac event and been through a hospital cardiac  rehabilitation phase. In addition, the study  aimed at verifying whether the patients’ inclusion in a primary  health care cardiac rehabilitation program allowed to increase the level of perceived self-efficacy.

The paper has a clear friendly structure (Introduction,  Materials and Methods, Results,  Discussion and Conclusions) and  is easy to read. The subject is interesting and useful. The introduction is clear and  sufficient  and the paper stands good in depth study design and methodology. The text is supplemented by  2 tables and  1 figure and enriched with 35 relevant references.

However there are several issues which certainly require  correction/supplements:

Major comments

1.       The Results section seems to be too concise and  not very clear, especially in the  second part when it comes to the main outcomes - a difference in means of 6.09 points in the General Scale of Self-Efficacy, lack of significant differences between the groups  in Hamilton scale and Beck Depression Inventory?  It requires a more detailed presentation,  perhaps in the form of tables?

2.       The Discussion section is too poor and needs to be expanded showing evidence of other similar studies in the field; at present in the paper there is only a general statement: ‘the results are consistent with the bibliographic evidence (1,6,23,25,27).’

3.       Limitations could be separated into a separate subsection.

Minor comments

1.       It would be worthwhile to add used scales (General Scale of Self-Efficacy, Hamilton Anxiety Rating Scale, Beck Depression Inventory etc.)) in the form of Annex to the paper.

Author Response

Major comments

Thank you very much for your comments and suggestions. The research team are grateful so to have the opportunity of to publish in this Journal.

The Results section seems to be too concise and not very clear, especially in the second part when it comes to the main outcomes - a difference in means of 6.09 points in the General Scale of Self-Efficacy, lack of significant differences between the groups in Hamilton scale and Beck Depression Inventory?  It requires a more detailed presentation, perhaps in the form of tables?

Thank you by your comments. The interpretation and the range of punctuation for the Self-Efficacy General Scale is explained in the text now. In addition, a new figure with the Scale and its items has been added.

6.09 points is the difference of mean´s punctuation between groups to study.

Finally, a new table is added to the text, including the analysis process.

The Discussion section is too poor and needs to be expanded showing evidence of other similar studies in the field; at present in the paper there is only a general statement: ‘the results are consistent with the bibliographic evidence (1,6,23,25,27).’

Thank you very much for your comments. The conclusions section has been extended according to your suggestions and to the references also.

Limitations could be separated into a separate subsection.

Thank you very much for your comments. The limitations section has been separated according to your suggestion.

Minor comments

It would be worthwhile to add used scales (General Scale of Self-Efficacy, Hamilton Anxiety Rating Scale, Beck Depression Inventory etc.)) in the form of Annex to the paper.

Thank you very much for your comments. A new figure with the General Scale of Self-Efficacy has been added to the text. All the scales or questionnaires have been validated to the Spanish population, and these has been referred in the text and with references.

Reviewer 2 Report

Cardiac rehabilitation program in primary care is a meaningful topic. While the introduction of the paper was well written, I have some suggestions to improve the other sections, especially the Methods and Results.

1.          In the Methods, the sentence “two patients assigned to the intervention group to one assigned to the control group” is a bit unclear. It is better to state it as a 2:1 ratio. Besides, the sample size calculation “44 subjects in each group” does not match with this ratio.

2.          Have all the scales including General Scale of Self-Efficacy, Salamanca screening questionnaire, the Hamilton Anxiety Rating Scale, and the Beck Depression Inventory been validated in Spanish?

3.          Table 1 and Table 2 look exactly the same. There is a mistake. Table 2 should be corrected.

4.          There was significant effect on self-efficacy but not depression and anxiety. This could be mentioned in the Abstract. What do 6.09 points mean in the General Scale of Self-Efficacy? The score range (minimum to maximum) of this scale should be described in the Methods as this is the key finding.

5.          The Discussion section is rather short. What is the impact of this study to cardiac rehabilitation programs in primary care settings in Spain and other countries? Is there room for improving the programs?

6.          Amendment of the following sentences is needed:

In the Abstract, the sentence “A clinical community trial, open, controlled, and randomised.” is fragmented.

In the Methods: a(perceived general self-efficacy)

Author Response

In the Methods, the sentence “two patients assigned to the intervention group to one assigned to the control group” is a bit unclear. It is better to state it as a 2:1 ratio. Besides, the sample size calculation “44 subjects in each group” does not match with this ratio.

Thank you very much for your comments. Your suggestion has been attended and included into the text (“Sample size, and sampling and randomization procedure” subsection).

2. Have all the scales including General Scale of Self-Efficacy, Salamanca screening questionnaire, the Hamilton Anxiety Rating Scale, and the Beck Depression Inventory been validated in Spanish?

Thank you very much for your comments. Yes. So is it. Furthermore. References of its validation have been added.

3. Table 1 and Table 2 look exactly the same. There is a mistake. Table 2 should be corrected.

Thank you very much for your comments. You are in true. There is a mistake. It has been modified.

4. There was significant effect on self-efficacy but not depression and anxiety. This could be mentioned in the Abstract. What do 6.09 points mean in the General Scale of Self-Efficacy? The score range (minimum to maximum) of this scale should be described in the Methods as this is the key finding.

Thank you by your comments. The interpretation and the range of punctuation for the Self-Efficacy General Scale is explained into the text now. In addition, a new figure with the Scale and its items has been added.

6.09 points is the difference of punctuation of means between groups to study. Finally, a new table is added to the text, including the analysis process

5. The Discussion section is rather short. What is the impact of this study to cardiac   rehabilitation programs in primary care settings in Spain and other countries? Is there room for improving the programs?

Thank you very much for your comments. The conclusions section has been extended according to your suggestions and to the references also. Specially, there is a reference in relation to a meta-analysis about it (reference number 39)

6. Amendment of the following sentences is needed:

In the Abstract, the sentence “A clinical community trial, open, controlled, and randomised.” is fragmented.

In the Methods: a(perceived general self-efficacy)

Thank you very much for your comments. Your suggestions have been take care of.

Round 2

Reviewer 1 Report

I am generally satisfied with all corrections and additions to the text  you have made.

I have only one remark regarding discussion: thank you for the broadening of the Discussion section - it is now much better. However the last sentence of the Discussion section is  not clear for me:

‘The results of this study are not consistent with data about women´s participation or dropout from  cardiac rehabilitation programs. (1, 6, 17, 23, 25, 27, 40)’.

What do you mean by this sentence? What about the references? They belong to the last sentence or to the whole Discussion section? Each reference should be referred to the concrete statement within the concrete sentence in the discussion e.g.  :

‘Psychoeducation resulted in  improved mental health in patients with coronary heart disease. There is evidence about benefits in case of this approach be performed as a complementary, effective, non-invasive nursing intervention to reduce psychological problems in these patients’.   There is no reference here.

‘There are results than also suggest that cardiac  rehabilitation programs that were successful on proximal targets (systolic blood pressure, smoking  behavior, physical exercise, emotional distress) were more effective on distal targets (cardiac  mortality and MI recurrences) than programs without success on proximal targets’.   There is no reference here as well, etc.

I have no more comments.

Author Response

Comments and Suggestions for Authors

I am generally satisfied with all corrections and additions to the text you have made.

I have only one remark regarding discussion: thank you for the broadening of the Discussion section - it is now much better. However the last sentence of the Discussion section is not clear for me:

‘The results of this study are not consistent with data about women´s participation or dropout from cardiac rehabilitation programs. (1, 6, 17, 23, 25, 27, 40)’.

What do you mean by this sentence? What about the references? They belong to the last sentence or to the whole Discussion section? Each reference should be referred to the concrete statement within the concrete sentence in the discussion e.g. :

‘Psychoeducation resulted in improved mental health in patients with coronary heart disease. There is evidence about benefits in case of this approach be performed as a complementary, effective, non-invasive nursing intervention to reduce psychological problems in these patients’.   There is no reference here.

‘There are results than also suggest that cardiac  rehabilitation programs that were successful on proximal targets (systolic blood pressure, smoking  behavior, physical exercise, emotional distress) were more effective on distal targets (cardiac  mortality and MI recurrences) than programs without success on proximal targets’.   There is no reference here as well, etc.

I have no more comments.

Thank you very much for your comments.

You are in true.  The references have been related with each sentence. The section has been modified.

Submission Date

09 May 2019

Date of this review

28 May 2019 15:05:57